# A Dihydropyridine Derivative as a Highly Selective Fluorometric Probe for Quantification of Au^3+^ Residue in Gold Nanoparticle Solution

**DOI:** 10.3390/s23010436

**Published:** 2022-12-30

**Authors:** Waroton Paisuwan, Mongkol Sukwattanasinitt, Mamoru Tobisu, Anawat Ajavakom

**Affiliations:** 1Department of Chemistry, Faculty of Science, Chulalongkorn University, Phyathai Road, Bangkok 10330, Thailand; 2Nanotec-CU Center of Excellence on Food and Agriculture, Department of Chemistry, Faculty of Science, Chulalongkorn University, Phyathai Road, Bangkok 10330, Thailand; 3Department of Applied Chemistry, Graduate School of Engineering, Osaka University, Osaka 565-0871, Japan; 4Innovative Catalysis Science Division, Institute for Open and Transdisciplinary Research Initiatives (ICS-OTRI), Osaka University, Osaka 565-0871, Japan

**Keywords:** gold(iii) ion detection, fluorescent sensor, fluorescence quenching, dihydropyridine, gold nanoparticles (AuNPs)

## Abstract

Novel dihydroquinoline derivatives (DHP and DHP-OH) were synthesized in one pot via a tandem trimerization-cyclization of methylpropiolate. DHP and DHP-OH possess strong blue fluorescence with high quantum efficiencies over 0.70 in aqueous media. DHP-OH displays a remarkable fluorescence quenching selectively to the presence of Au3+ through the oxidation of dihydropyridine to pyridinium ion as confirmed by NMR and HRMS. DHP-OH was used to demonstrate the quantitative analysis of Au3+ in water samples with the limit of detection of 33 ppb and excellent recovery (>95%). This fluorescent probe was also applied for the determination of Au3^+^ residue in the gold nanoparticle solution and a paper-based sensing strip for the on-site detection of Au3^+^.

## 1. Introduction

Gold, a rare and precious metal, has been extensively used for various applications such as an anti-aging agent in cosmetics [1], a highly efficient catalyst in organometallic chemistry [2,3,4,5], a highly electrical conductor in the electronic industry [6], a nanoparticle substrate [7,8,9], and a bioactive ingredient in pharmaceuticals [10,11]. Recently, gold ions were used as a therapeutic agent that shows excellent biocompatibility in medication for treatment in a wide variety of diseases including rheumatic arthritis, asthma, cancer, AIDS, bronchial asthma, malaria and brain lesions [12,13,14,15,16,17]. Nevertheless, excessive exposure of the human body to gold(III) ions (Au^3+^) can be potentially toxic and cause damage to several organs including kidneys, lungs, spleen, liver and the peripheral nervous system [14,16,18]. Hence, the quantification of Au^3+^ is important for industrial waste management, environmental monitoring, and medical diagnosis.

Typically, there are several techniques for the Au^3+^ quantification including atomic absorption spectroscopy (AAS), inductively coupled plasma mass spectroscopy (ICP-MS), inductively coupled plasma atomic emission spectroscopy (ICP-AES), electrochemical analysis, X-ray fluorescence spectrometry, neutron activation analysis, and UV–vis spectroscopy [19,20,21,22,23,24]. Fluorescence spectroscopy is one of the effective detecting techniques due to its high sensitivity, specificity, and low cost with good visualizing capability. On top of that, a paper-based strip fabricated with selective fluorescent probe can be potentially combined with this technique to be a portable test kit, which is suitable for on-site analysis [25,26,27,28,29,30].

Notably, various fluorescent probes for a Au^3+^ sensing system have been developed from derivatives of rhodamine [31], julolidine [32], naphthalimide [33], spirobifluorene [34], and boron dipyrromethene (BODIPY) [35,36]. However, these compounds were prepared from complicated and expensive starting chemicals. Recently, we have found that 1,4-dihydropyridine derivatives can be readily prepared from simple propiolate ester and could be used for Cu^2+^ [37] and Hg^2+^ [38] detection. We are therefore interested in further development of 1,4-dihydropyridine derivatives as novel fluorescent probes for the detection of highly oxidative metal ions such as Au^3+^ in aqueous solution. Two dihydropyridine derivatives, DHP and DHP-OH, were conceptually designed as novel hydrophilic fluorescent sensors (Figure 1). DHP-OH was applied on the quantification of Au^3+^ in water samples from various sources and Au^3+^ residue remained in the preparation of gold nanoparticles (AuNPs). Furthermore, the development of a DHP-OH probe as a paper-based sensing strip for on-site analysis of Au^3+^ is also reported herein.

## 2. Materials and Methods

### 2.1. Chemicals

All reagents were purchased from Sigma-Aldrich Co. (Aldrich) and Tokyo chemical industrial Co. Ltd. (TCI), such as ammonium acetate (NH_4_OAc), 2-aminoethan-1-ol, methylpropiolate, piperazine, chlorotrimethylsilane (TMSCl), anhydrous sodium sulfate, sodium chloride, 2-Amino-2-(hydroxymethyl)propane-1,3-diol hydrochloride (Tris-HCl), and were used without further purification. Solvents used for the reaction included dimethyl sulfoxide (DMSO) and water (H_2_O). Solvents for other uses (e.g., screening experiments, extraction, and chromatography), acetonitrile (CH_3_CN), deionized H_2_O, dichloromethane (CH_2_Cl_2_), ethanol (EtOH), ethyl acetate (EtOAc), hexane, and tetrahydrofuran (THF) were commercial grade and distilled prior to use. Column chromatography was performed on Merck silica gel 60 (70–230 mesh). Milli-Q water was used to prepare stock metal ions for UV-vis and fluorescence experiments. Metal ions were prepared from their commercially available nitrate salts, except for mercury(II), antimony(III), and iron(II), which were in the form of acetate salts, and gold(III) in the form of 30%wt HAuCl_4_ solution. 2-[4-(2-Hydroxyethyl)piperazin-1-yl]ethane-1-sulfonic acid (HEPES) was used to reduce gold(III) to AuNPs.

### 2.2. Preparation of Stock Solutions of DHP, DHP-OH, and Metal Ions

DMSO was used to dissolve DHP and DHP-OH for the preparation of 10 mM DHP and DHP-OH stock solutions. There were 26 types of stock metal ions consisting of Li^+^, Na^+^, K^+^, Rb^+^, Cs^+^, Mg^2+^, Ca^2+^, Sr^2+,^ Ba^2+^, Al^3+^, Ga^3+^, Bi^3+^, Fe^3+^, Cr^3+^, Fe^2+^, Zn^2+^, Cd^2+^, Pb^2+^, Co^2+^, Cu^2+^, Ni^2+^, Hg^2+^, Mn^2+^, Ag^+^, Sb^3+^, and Au^3+^. All metal salts prepared from their nitrate salts, except for Hg(OAc)_2_, Sb(OAc)_3_, Fe(OAc)_2_, and HAuCl_4_ were dissolved in Milli-Q water to produce 10 mM aqueous solutions. Various concentrations were then obtained by diluting these stock solutions with appropriate solvents.

### 2.3. Instruments

High-resolution mass spectrometry (HRMS) involved the use of an electrospray ionization mass spectrometer (MicroTOF, Bruker Daltonics, Billerica, USA). ^1^H NMR and ^13^C NMR spectra were recorded on a JEOL JNM-ECZR (Jeol, USA) at 500 MHz and 125 MHz, respectively. The UV-visible spectra were obtained on a Varian Cary 50 UV-vis spectrophotometer (Varian, USA). The emission spectra were obtained from a Carry Eclipse fluorescence Spectrophotometer (Agilent Technologies). IR spectra results were acquired from the ALPHA II Compact Fourier-transform infrared (FTIR) Spectrometer (Bruker, Billerica, USA). The pH values of the experimental solution were measured from an Ohaus pH meter, and the UV (λ_ex_ = 312 nm) and blacklight (λ_ex_ = 365 nm) sources from standard transilluminator TCP-20.LM V1 UV 365/312 nm (Vilber Lourmat, Collégien, France) were used for naked eye observations in the sensing experiments. The optical and morphological characterization of AuNPs were obtained from the Dynamic Light Scattering (DLS) of a Zetasizer Nano ZSP (Malvern, UK) with a 1.00 cm quartz cell at room temperature and Transmission Electron Microscopy (TEM) of a JEM-1400 electron microscope under an accelerating voltage of 120 kV (Jeol, USA).

### 2.4. General Procedure for Fluorescence Sensing on Solid Phase

In order to develop a paper-based sensing strip for Au^3+^ detection, wax-printed filter paper strip was prepared by a wax printer (Xerox ColorQube 8870, Connecticut, USA) to create circular patterns (diameter at 7 mm) on filter paper, which was designed by Adobe Illustrator 2022. Heating at 110 °C for 30 s on a hotplate was then performed to melt the wax and create the final hydrophobic patterns (diameter of around 5 mm). Then, the 10 mM DHP-OH solution in DMSO was dropped onto this paper strip (1.00 µL) and allowed to air-dry overnight. The prepared solid state of DHP-OH (5 nmol) showed the bright blue fluorescence under blacklight. The paper-based sensing strip was dropped with 5 nmol of each metal ion solution using Milli-Q water as a blank.

### 2.5. Preparation of Water Samples

Drinking water, tap water, and rainwater were selected for the sample tests. The crude water samples were filtered through a microfiltration membrane (0.45 μm) before use. Au^3+^ (3.00 ppm) was spiked into the water samples prior to the experiments. The mixture of spiked water samples and DHP-OH were prepared to obtain the final concentration of DHP-OH (3.13 ppm) in a 10 mM Tris-HCl buffer solution at pH 6.0. The fluorescence spectra of these mixtures were measured after preparation for overnight to assure the completion of the reaction.

### 2.6. Preparation of AuNPs

Gold nanoparticles were synthesized by the different molar ratios of HAuCl_4_/HEPES (1:0.25, 1:0.50, 1:1, 1:2, 1:4, 1:8, 1:10, 1:15, and 1:20) at a total volume of 1000 µL in a polypropylene micro-centrifuge tube. As a detailed synthetic procedure, 10 mM HAuCl_4_ solution (10 µL) was first diluted with Milli-Q water followed by the addition of 10 mM HEPES solution (2.5, 5, 10, 20, 40, 50, 100, 150, and 200 µL) and the mixture was kept for overnight. These mixtures were optically and morphologically investigated by UV-vis absorption, DLS, and TEM experiments. To determine the amount of Au^3+^ residue from the AuNP preparation with the DHP-OH probe, each mixture (200 µL) was diluted with 10 mM Tris-HCl solution at pH 6.0 (799 µL). Then, the solution was added with 10 mM DHP-OH (1 µL) and kept for another overnight before measuring the fluorescence intensity.

## 3. Results and Discussion

### 3.1. Synthesis and Characterization

The synthesis of dihydropyridine derivatives was executed in one pot via a tandem trimerization-cyclization of methylpropiolate with an amine using piperazine and TMSCl as cocatalysts [39] (Figure 1). Ammonium acetate and 2-aminoethan-1-ol were used for the preparation of DHP and DHP-OH, respectively, in good yields. The compounds were confirmed by ^1^H NMR, ^13^C NMR, ATR-FTIR, and HRMS data (Appendix A). Both DHP and DHP-OH exhibit bright blue fluorescence in aqueous solution under the blacklight (λ = 365).

The absorption and emission spectra of DHP and DHP-OH were explored in aqueous solution (Figure 2). The maximum absorption and emission wavelengths (λ_ab_ and λ_em_) are at 358 and 431 nm, respectively for DHP as well as 369 and 444 nm, respectively for DHP-OH. Moreover, the molar extinction coefficients (Ɛ) of DHP and DHP-OH investigated in various concentrations and were calculated to be 3.0 × 10^3^ and 3.3 × 10^3^ M^−1^cm^−1^, respectively. The high fluorescence quantum efficiency (Φ_f_) of DHP and DHP-OH could be observed to be 0.73 and 0.89, respectively by using quinine sulfate in 0.1 M H_2_SO_4_ as a reference (Appendix A).

### 3.2. pH Effect and Time Dependence

The study of pH effect on the fluorescence properties of DHP and DHP-OH was performed in the pH range of 3.0–11.0 (Appendix A). Although DHP and DHP-OH similarly exhibit the high fluorescence intensity at all pH ranges demonstrating pH tolerance properties, the sensing region of Au^3+^ by both fluorescent probes was limited from pH 3.0 to 6.0. Tris-HCl buffer was needed for the pH control in this sensing system. The results of time dependent fluorescence quenching profile at 10 equivalents of Au^3+^ indicate that the fluorescence emission of both compounds was completely quenched within 40 min for DHP-OH, whereas more than 60 min were required in the case of DHP (Appendix A). DHP-OH was thus used for further metal ion sensing studies due to its superior fluorescence properties and faster response. As to the reason for faster response of DHP-OH than that of DHP, the hydroxyl group of DHP-OH may promote the reaction by a chelating effect with AuCl_3_.

### 3.3. Metal Ion Sensing Properties

The sensing properties of DHP-OH probe were tested with 26 metal ions in the solution of Tris-HCl (pH 6.0) at ambient temperature for 40 min. Interestingly, it showed a highly selective fluorescence quenching with Au^3+^ that was easily observable by the naked eye under blacklight (Figure 3).

For quantitative analysis, the fluorescence titration of DHP-OH with Au^3+^ was studied (Figure 4). When the amount of Au^3+^ was increased, the fluorescence emission spectra gradually decreased. A linear plot between the fluorescence intensity at 444 nm and Au^3+^ concentration was obtained in the range of 0–3 ppm. The limit of detection (LOD), depending on 3SD/K (where SD is the standard deviation of the blank and K is the slope of the plot), could be estimated to 33 ppb (167 nM) with R^2^ = 0.9911 (inset of Figure 4). These results indicate that the DHP-OH probe is comparable to other sensitive fluorescent sensors for Au^3+^ in aqueous media (Appendix A) [31,32,34,35,36,40,41,42].

Subsequently, the competitive experiments were carried out to investigate the interference from foreign metal ions (25 types) including Li^+^, Na^+^, K^+^, Rb^+^, Cs^+^, Mg^2+^, Ca^2+^, Sr^2+,^ Ba^2+^, Al^3+^, Ga^3+^, Bi^3+^, Fe^3+^, Cr^3+^, Fe^2+^, Zn^2+^, Cd^2+^, Pb^2+^, Co^2+^, Cu^2+^, Ni^2+^, Hg^2+^, Mn^2+^, Ag^+^, and Sb^3+^ (Figure 5). The fluorescence quenching signal in the presence and the absence of foreign metal ions were relatively at the same level except for Fe^3+^ and Cu^2+^, which showed significantly stronger quenching. These results demonstrate low interference from foreign metal ions, especially for common metal ions such as alkaline and alkaline earth metal ions.

To investigate the source of the fluorescence quenching of DHP-OH by Au^3+^, UV-vis experiments were conducted. First, the absorption band of the Au^3+^ solution does not overlap with that of DHP-OH, which confirms the absence of a competitive absorption process (Appendix A). Furthermore, we observed the time dependent change of the DHP-OH band upon the addition of Au^3+^ (Figure 6). The peaks at 320 and 369 nm gradually decreased, while the new peak at 275 nm increased with an isosbestic point at 285 nm. These results are in good agreement with our previous observation of the oxidation of dihydropyridine to pyridinium ion [38].

### 3.4. Sensing Mechanism

To gain insight into the fluorescence quenching mechanism of DHP-OH by Au^3+^, NMR and HRMS experiments were executed. According to the ^1^H NMR spectra of DHP-OH responding to an excess amount of Au^3+^ (Figure 7), the H_a_, H_b_, H_c_, H_e_, H_f_, and H_g_ signals were downfield shifted and the H_d_ signal completely disappeared, corresponding to the conversion of the dihydropyridine ring into an aromatic pyridinium ring [38]. In addition, the ^13^C NMR and HRMS results confirmed the formation of pyridinium ring with very down field aromatic peaks in the range of 155–149 ppm [43,44] and the molecular ion peak at m/z = 312.1077, respectively (Appendix A).

Although the sensing mechanism of DHP-OH involves the oxidation of dihydropyridine to pyridinium ion, its Au^3+^ selectivity is different from the Hg^2+^ selectivity found in our previously reported aromatic substituted dihydropyridine. We believe that the difference in selectivity is due to two parameters: 1) Au^3+^ is the harder Lewis acid, and 2) the *N*-aliphatic substituent does not promote the formation of an amine radical cation. Therefore, the sensing mechanism in the present discovery may start with the binding between the Au^3+^ Lewis acid and the DHP-OH Lewis base (Figure 2). The formation of the Lewis acid-base complex is also evidenced by a new transient UV-vis absorption at 320 nm after the addition of Au^3+^ (compare Figure 6 with Figure 2). The subsequent reductive elimination of the Au^3+^ complex generates the non-fluorescent pyridinium ion along with Au^+^ species.

### 3.5. Application on Paper-Based Sensing Strip

To develop a fluorescent probe for on-site analysis, a paper-based sensing strip is alternatively chosen as a probe platform due to its low cost and easy portability. Therefore, we deposited DHP-OH on a wax-printed filter paper strip for the metal ion detection. The bright blue fluorescence emission was dramatically quenched by Au^3+^ within 5 min by naked-eye observation (Figure 8a). Apparently, the rate of Au^3+^ sensing on the paper-based sensing strip was faster than that of the solution system, possibly due to the reduced solvation effect towards Au^3+^. Naked eye detection of this paper-based strip was determined to be 0.2 nmol (39 ng) of Au^3+^ (Figure 8b).

### 3.6. Quantitative Analysis of Au^3+^ in Water Samples

To demonstrate a quantitative analysis of Au^3+^ in water samples from various sources including drinking water, tap water, and rainwater, the small amount of Au^3+^ stock solution was spiked in each sample and diluted with Tris-HCl solution to obtain the final concentration at 3.00 ppm. The fluorescence intensity of water samples was determined by a linear equation (inset of Figure 4). The data in Table 1 show that the DHP-OH probe could be practically employed for Au^3+^ detection in water samples with excellent recoveries of 96–98%.

### 3.7. Quantitative Analysis of Au^3+^ Residue in the AuNP Solution

Currently, AuNPs have been extensively used in many applications. The determination of Au^3+^ residue in the AuNP solution can thus be important for evaluating toxicity in the applications and cost effectiveness in the preparations of AuNPs. However, no analytical technique was reported for determination of Au^3+^ residue in the AuNP solution. Therefore, we decided to utilize a DHP-OH probe to analyze the concentration of Au^3+^ residue in the AuNP solution prepared from the reduction of HAuCl_4_ with various concentrations of HEPES [45,46]. The results indicate that the Au^3+^ residue decreased with the increased amount of HEPES used, which is in a good agreement with the increase of AuNP absorption at 547 nm and the DLS count rate of the nanoparticles formed (Figure 9). The UV-vis absorption spectra, DLS spectra, and TEM images clearly confirmed the presence and morphology of AuNPs formed in the preparation (Appendix A). It indicates that the DHP-OH probe could be applied for the determination of Au^3+^ residue in AuNP solution.

## 4. Conclusions

Two dihydropyridine derivatives (DHP and DHP-OH) were successfully developed as hydrophilic fluorescent sensors for the Au^3+^ sensing system. DHP-OH shows highly selective fluorescence quenching towards Au^3+^ in aqueous media with a detection limit of 33 ppb (167 nM). The sensing mechanism was proven to be an Au^3+^ mediated oxidation of dihydropyridine to pyridinium ion. Furthermore, the development of a DHP-OH probe-based paper strip was magnificently achieved as an on-site kit for Au^3+^ detection. The quantification of Au^3+^ ions in water samples was accomplished with excellent recoveries. Moreover, the difficulties on the confirmation of Au^3+^ residue in AuNP preparation could be solved by using our optimized gold sensing system.

## 5. Experimental Section

### General Synthetic Procedure of DHP and DHP-OH

Ammonium acetate (500 mg, 6.49 mmol) or 2-aminoethan-1-ol (392 µL, 6.49 mmol) were dissolved in 10 mL of H_2_O:DMSO (1:1, *v*/*v*) in an ace pressure tube. Methylpropiolate (1731 µL, 3 equivalents), piperazine (111 mg, 0.2 equivalents), and TMSCl (165 µL, 0.2 equivalents) were added into the reaction solution. The mixture was heated and stirred at 120 ^°^C overnight. The reaction mixture was evaporated under reduced pressure, washed with saturated brine solution, and extracted with CH_2_Cl_2_. The organic fraction was dried over anhydrous Na_2_SO_4_ and evaporated under vacuum pressure. The crude product was purified by column chromatography (Silica gel, Hexane:EtOAc = 1:1, *v*/*v*) to obtain DHP or DHP-OH as a yellow solid in the yield of 72% or 83%.


**Dimethyl 4-(2-methoxy-2-oxoethyl)-1,4-dihydropyridine-3,5-dicarboxylate (DHP):**


^1^H NMR (500 MHz, acetone**-***d_6_*) δ (ppm): 8.38 (br s, 1H), 7.38 (s, 1H), 4.16 (t, *J* = 5.6 Hz, 1H), 3.67 (s, 6H), 3.52 (s, 3H), 2.34 (d, *J* = 5.6 Hz, 2H); ^13^C NMR (125 MHz, acetone**-***d_6_*) δ (ppm): 172, 168, 137, 105, 513, 51.2, 41.4, 30.5. HRMS (ESI): calculated for [M+Na]^+^ 292.0792, found 292.0806.


**Dimethyl 1-(2-hydroxyethyl)-4-(2-methoxy-2-oxoethyl)-1,4-dihydropyridine-3,5-dicarboxylate (DHP-OH):**


^1^H NMR (500 MHz, acetone**-***d_6_*) δ (ppm): 7.30 (s, 2H), 4.11 (t, *J* = 5.2 Hz, 1H), 3.74 (t, *J* = 5.4 Hz, 2H), 3.67 (s, 6H), 3.61 (t, *J* = 5.2 Hz, 2H), 3.54 (s, 3H), 2.37 (d, *J* = 5.2 Hz, 2H); ^13^C NMR (125 MHz, acetone**-***d_6_*) δ (ppm): 172, 168, 141, 106, 62.1, 57.7, 51.5, 51.4, 41.3, 30.4. HRMS (ESI): calculated for [M+Na]^+^ 336.1059, found 336.1034.

## Data Availability

All data generated or analysed during this study are included in this published article and its Appendix A.

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
