# Peer review of "A Dihydropyridine Derivative as a Highly Selective Fluorometric Probe for Quantification of Au3+ Residue in Gold Nanoparticle Solution"

_sensors, 2022, doi:10.3390/s23010436_

Round 1
Reviewer 1 Report
Dear authors,
I enjoyed reading your article. It is a clear and well-written article.
I have only one point that should be raised and discussed in your article.
The AuNP is known to be a catalyzer for many types of reactions (one example, https://www.frontiersin.org/articles/10.3389/fchem.2019.00702/full). Your solutions are not deoxygenated. Are you sure that your dyes are not oxidized by AuNP in the presence of oxygen?
In addition, can you please discuss the effects of oxygen, are there any effects?
best wishes
Author Response
Q: Your solutions are not deoxygenated. Are you sure that your dyes are not oxidized by AuNP in the presence of oxygen? In addition, can you please discuss the effects of oxygen, are there any effects?
A: In the exemplified case of the gold catalyzed oxidation of alcohols, we believed that its use is limited under the method dealing with immobilization of colloidal particles (COL) with narrow nanoparticle size distribution (around 3–6 nm). We are sure that our AuNP system cannot initialize the oxidation seen in such conditions, because our diameters of NPs are much bigger (60-100 nm) and does not have the activating effect from the immobilization with microporous carbon.
In addition, according to our previous work as shown in Table 4 (Entry 6) (https://doi.org/10.1016/j.sbsr.2021.100470.), 1,4-dihydropyridine (DHP) derivatives are stable under oxygen (O2) condition; however, they can be oxidized by using a harsh conditions of H2O2 combined with catalytic Cu2+. This fact supports the high stability of such DHP derivatives, and therefore, we really believe that our DHP-OH sensor cannot be oxidized by AuNP with/without oxygen. Again, the results of the sensing properties of our DHP-OH toward the metal ions showed the absolute selectivity to Au3+, together with the results of Figure 9 demonstrating the fluorescent intensity of DHP-OH decreased (more DHP-OH were oxidized) when the Au3+ residue increased (less AuNPs were produced) also directly lead us to this conclusion.
Moreover, O2 does not normally involve in the process of the AuNP formation. And we have never seen such the use of the deoxygenated solution for the nanoparticle formation process as well. Therefore, we also believe that there are no other effects of O2 in our system.
Reviewer 2 Report
The authors developed two dihydropyridine derivatives for fluorescence sensing of Au3+. The sensing mechanism was well studied. The quantification of Au3+ ions in water samples was accomplished with excellent recoveries. Paper-based sensing strips for on-site detection of Au3+ were also constructed. I recommend its acceptance in Catalysts, if the authors address the following issues.
1. In figure 2. The y-axis is incorrectly labeled.
2. The detection limit should not start with 0. What is the lowest detectable concentration?
3. Besides the interference experiment shown in Figure 5, the authors are suggested to carry out a competitive experiment, in which only foreign metal ions are present without Au3+.
Author Response
Q1. In figure 2. The y-axis is incorrectly labeled.
Thank you for your kind correction. We have accordingly, changed the y-axis label from “normalized spectra” into “normalized intensity”.
Q2. The detection limit should not start with 0. What is the lowest detectable concentration?
We understood the point that the reviewer concerns, because the “Au3+ concentration was obtained in the range of 0-3.00 ppm” seen in the paragraph explaining Fig. 4 seemed to be awkward. Therefore, we did alter this phrase into the real result which is “Au3+ concentration was obtained in the range of 0.11-3.00 ppm”. This “0.11 ppm” was calculated from the method for the determination of LOQ (lowest detectable concentration). In details, the 10SD/K value was used instead of the conventional LOD one (3SD/K value) resulting in the lowest detectable concentration of 109.3 ppb. Hopefully we are not misunderstanding the reviewer’s question, and it does certainly answer the reviewer’s query.
Q3. Besides the interference experiment shown in Figure 5, the authors are suggested to carry out a competitive experiment, in which only foreign metal ions are present without Au3+.
Thank you for your suggestion. We believe that the sensing experiment executed in Figure 3 can fulfill reviewer’s question, as they were done in the absence of Au3+ with 10-times higher concentration of other metal ions than those used in the interfering experiment (Figure 5).